# The Impact of the Quality of Tap Water and the Properties of Installation Materials on the Formation of Biofilms

**Dorota Papciak [1], Barbara Tchórzewska-Cieślak [2], Andżelika Domoń [1,*], Anna Wojtuś [1], Jakub Żywiec [2] and Janusz Konkol [3]**

[1] Department of Water Purification and Protection, Faculty of Civil, Environmental Engineering and Architecture, Rzeszow University of Technology, Al. Powstancow Warszawy 6, 35-959 Rzeszow, Poland; dpapciak@prz.edu.pl (D.P.); annawojtusss@gmail.com (A.W.)

[2] Department of Water Supply and Sewerage Systems, Faculty of Civil, Environmental Engineering and Architecture, Rzeszow University of Technology, Al. Powstancow Warszawy 6, 35-959 Rzeszow, Poland; cbarbara@prz.edu.pl (B.T.-C.); j.zywiec@prz.edu.pl (J.Ż.)

[3] Department of Materials Engineering and Technology of Building, Faculty of Civil, Environmental Engineering and Architecture, Rzeszow University of Technology, Al. Powstancow Warszawy 6, 35-959 Rzeszow, Poland; janusz.konkol@prz.edu.pl

\* Correspondence: adomon@prz.edu.pl; Tel.: +48-17-865-1949

**Abstract:** The article presents changes in the quality of tap water depending on time spent in installation and its impact on the creation of biofilms on various materials (polyethylene (PE), polyvinyl chloride (PVC), chrome-nickel steel and galvanized steel). For the first time, quantitative analyses of biofilm were performed using methods such as: Adenosine 5′-triphosphate (ATP) measurement, flow cytometry, heterotrophic plate count and using fractographical parameters. In the water, after leaving the experimental installation, the increase of turbidity, content of organic compounds, nitrites and nitrates was found, as well as the decrease in the content of chlorine compounds, dissolved oxygen and phosphorus compounds. There was an increase in the number of mesophilic and psychrophilic bacteria. In addition, the presence of *Escherichia coli* was also found. The analysis of the quantitative determination of microorganisms in a biofilm indicates that galvanized steel is the most susceptible material for the adhesion of microorganisms. These results were also confirmed by the analysis of the biofilm morphology. The roughness profile, the thickness of the biofilm layer can be estimated at about 300 μm on galvanized steel.

**Keywords:** biofilm; tap water; water supply system

## 1. Introduction

Water supply systems in residential buildings, public utilities or industrial plants are made of various materials [1]. The type of material and the quality of tap water are the most important factors affecting the risk of losing water safety reaching the consumer [2]. Therefore, the assessment of risks related to internal water supply installations, including hazards related to products and materials in contact with drinking water is extremely important [3]. A particular health risk is caused by the emergence of biofilm and increased colonization of organisms on the surface of water pipes. The intensity of biofilm growth in a water distribution system depends on numerous factors such as content of biogenic compounds in water injected into a network, amount of disinfectant, temperature, hydrodynamic conditions and the type of material from which the water conduits are made [4–8]. The chemical composition of the water supply system material, as well as its properties,

that is, porousness and susceptibility to corrosion, are regarded as one of the main causes of increased colonization of microorganisms. The multitude of factors influencing the formation and development of biofilm makes prevention of the phenomenon very complex [9–11].

Plastics and polyester resins in tap water distribution networks are replacing cast-iron and galvanized steel, however; pipes and elements made of copper or chromium-nickel steel are used as well. These materials are characterized by high resistance to corrosion and low surface porousness. Due to their ever-reducing price and valuable properties, they can become genuine competition for traditional materials used in water supply systems (such as polyethylene (PE), and polyvinyl chloride (PVC)). Currently, cast-iron and concrete are used decidedly less often to build water-pipe networks than just a few years ago, however; numerous sections of networks made of these materials are still in exploitation [12].

Synthetic materials characterized by low porousness were expected to eliminate corrosion and decrease the risk of secondary water pollution. The research conducted thus far shows that biofilm forms on all water supply system materials, but each of them creates different conditions for duplication and adhesion of microorganisms in the form of biofilm [13].

According to the prevailing literature on the issue, plastics can support the formation of biofilm, however, the growth of microorganisms on the surface of plastic pipes is usually the same or lower than in the case of iron, steel, or concrete [4,7].

Contact between water and the internal surface of the water conduits may lead to, i.e., corrosion or ageing of materials, eluting chemical substances, and to the formation of biofilm on their surfaces. Metals from brass elements, e.g., joints, can also be transferred into the water. This applies particularly to lead, which presents a serious risk to pregnant women and children below six years of age [1].

Phosphorus and carbon, which are nutrients for microorganisms, can transfer to water when materials come into contact with them in the form of microbiologically available phosphorus (MAP) [5] or available organic carbon (AOC) [14], speeding up biofilm formation.

Despite the possible influence of material on biofilm formation [15–17], in research conducted on actual conduit sections after a year of contact with tap water, this dependency has not been noted [1]. It was determined, however, that the biofilm formation process was influenced by water temperature and flow conditions [1].

Some researchers state that type of material influences biofilm formation, but only at its initial stage. According to the research, the small difference in the number of microorganisms populating internal surfaces of PE and copper pipes after 21 and 200 days shows the lack of dependency between material type and intensity of biofilm growth after a longer time period in contact with the medium [5].

None of the so far examined materials allows the complete elimination of biofilm formation in tap water distribution networks. It is, therefore, noteworthy that, due to their structure, both plastics and corroding materials create different opportunities for the formation of biofilm [5,13,16,18].

The research so far has focused on specific materials and selected different conditions of biofilm formation on material surfaces [1,4,5,18,19], experiment time duration [1,4,5,19,20], biofilm detachment method for analysis [4,5,18–20] and biofilm quantification methods [1,4,5,18,19]. Therefore, this article presents an evaluation method of the susceptibility of water supply system materials to biofilm formation in conditions close to those of actual water supply systems.

An additional possibility of analyzing biofilm morphology is provided by the use of fractal analysis together with the appropriate tools of fractal geometry [21] used to quantify the roughness of any structure. The biofilm research in such a wide range has not been conducted so far.

The aim of the study was to assess changes in water quality and biological stability depending on the time spent in the distribution system and to determine the susceptibility of materials to the adhesion of microorganisms.

## 2. Materials and Methods

### 2.1. Subject of Study

The experimental water supply system consisted of three main parts: A tap water connection, circulation in the experimental system, and a discharge of water from the system (Figure 1). Two ball valves and a water meter measuring the fresh tap water flow were installed on the connection conduit. The experimental water supply system was a closed circuit made of PVC DN (Diameter Nominal) 32 pipes, in which circulation was ensured by an installed pump. The circuit was equipped with a separate water meter, drain and vent lines, flow decrease installation, and disinfectant injection point. The samples of examined materials (PVC, PE, chromium-nickel steel, galvanized steel) attached with a stainless metal rod to PVC plugs along with an O-ring rubber seal, were mounted on specially modified tees separated with ball valves (Figure 2). The discharge of water from the system was obtained via a combination of drain and vent lines. Water flow speed in the experimental system equaled 0.3 m/h. A collection of samples for physical, chemical, and microbiological tests was via a tap with a cable, installed on the conduit discharging water from the system to the sewage system.

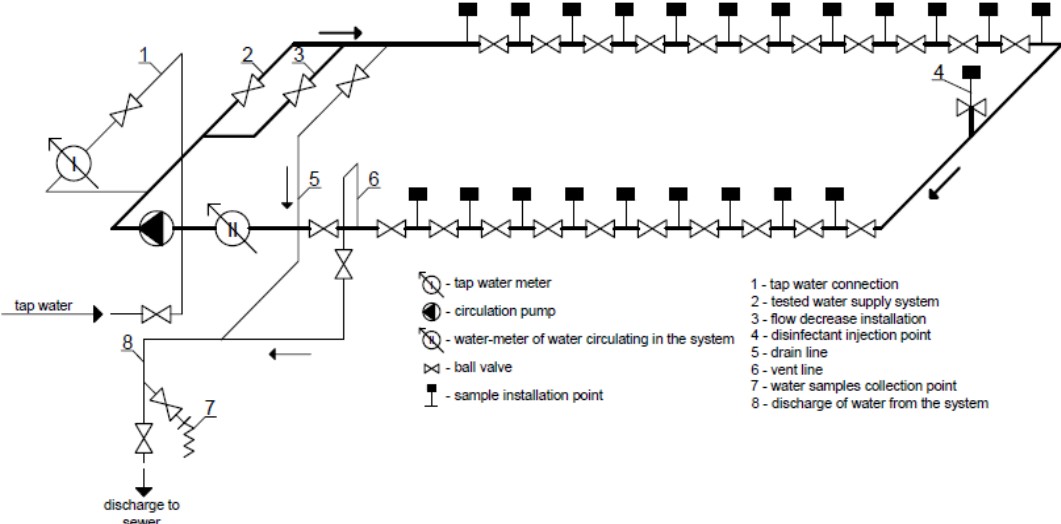

**Figure 1.** The scheme of the experimental installation.

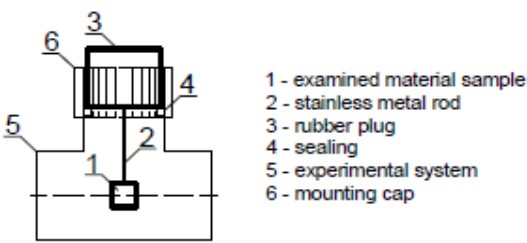

**Figure 2.** The scheme of the tested sample.

In the experimental installation, all analyzed installation materials were installed. Five coupons were prepared for each material, placed in the following order: PE, PVC, chromium-nickel steel, galvanized steel. Fresh tap water was introduced into the installation once a day, and then the water circulated in the closed circuit in the installation for 24 hours. Before starting the tests, the installation together with the coupons was disinfected with 15% sodium hypochlorite.

The building with the installation was located 5 km from the water treatment plant.

### 2.2. Water in the Experimental Installation

The experimental system was supplied with surface water which had been treated using the technology presented in Figure 3 and met the quality requirements of water intended for human consumption. Tests on quality changes in the water leaving and supplying the experimental system were conducted once a week for 6 months. Selected physical, chemical, and bacteriological parameters of water supplying and leaving the experimental system were estimated using the methods presented in Table 1. The tests were carried out in accordance with the applicable test procedures and the manufacturer's instructions were followed.

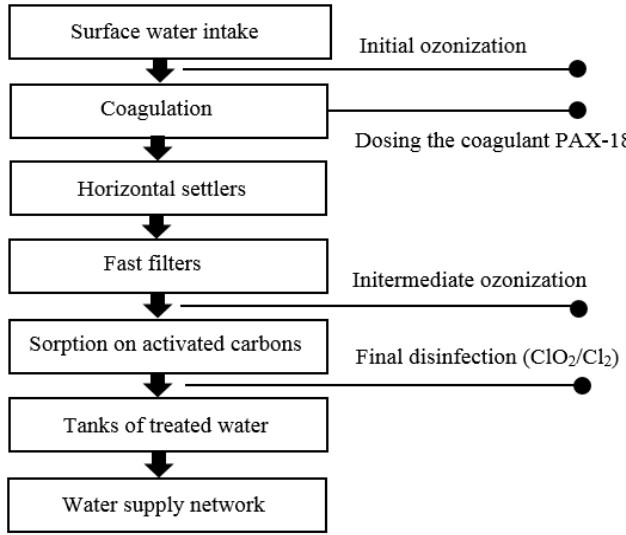

**Figure 3.** Water treatment technology.

**Table 1.** Analytical methods and standards used in experiment.

| Parameter | Analytical Method/Standard |
|---|---|
| pH | Multifunction meter CX-505 (Elmetron, Poland) |
| Temperature | Multifunction meter CX-505 (Elmetron, Poland) |
| Conductivity | Multifunction meter CX-505 (Elmetron, Poland) |
| Turbidity | 2100P ISO turbidimeter (Hach, Germany) |
| Oxidizability | The permanganate method with $KMnO_4$ (according PN EN ISO 8467:2001) |
| Total organic carbon (TOC) | TOC analyzer Sievers 5310 C (SUEZ, Boulder, CO, USA) |
| UV absorbance | Spectrophotometric method using Hach–Lange DR 500 spectrophotometer (Germany) |
| Dissolved oxygen | Electrochemical method using a Hach–Lange oxygen probe (Germany) |
| Ammonium nitrogen | Spectrophotometric method 8155 (sachet tests—amonia salicylate (1) and cyanurate (2)) using Hach-Lange DR 500 spectrophotometer (Germany) |
| Nitrite nitrogen | Colorimetric method by Nitrite Test Merck 1.14408 (Germany) |
| Nitrate nitrogen | Spectrophotometric method 8039 (sachet tests—NitraVer5) using Hach–Lange DR 500 spectrophotometer (Germany) |
| Phosphates | Spectrophotometric method 8048 (sachet tests—PhosVer3) using Hach–Lange DR 500 spectrophotometer (Germany) |
| Total and free chlorine | Spectrophotometric method 8167 and 8021 (sachet tests—DPD reagent) using Hach-Lange DR 500 spectrophotometer (Germany) |
| The total number of psychrophilic bacteria (at 22 °C) and mesophilic bacteria (at 37 °C) | Heterotrophic plate count (HPC) method using R2A Agar (CM0906) manufactured by Oxoid Thermo Scientific (UK) (incubation 7 days) and an A Agar (P-0231) manufactured by BTL Sp. z o.o. Department of Enzymes and Peptones (Poland) (incubation 2 day-mesophilic bacteria and 3 day psychrophilic bacteria) |
| *Escherichia coli* | Membrane filtration procedure using Endo agar WG ISO 9308-1 (BTL, Poland) |

　　　Biodegradable dissolved organic carbon (BDOC) content was calculated on the basis of data published by Wolska, who determined that in the case of surface waters, the dominant fraction of dissolved organic carbon (DOC) was the non-biodegradable fraction and equaled 90% of total organic carbon (TOC), while BDOC content equaled ca. 10.6% of DOC [11].

*2.3. The Susceptibility of Materials to the Formation of Biofilms in the Experimental Installation*

　　　The assessment of susceptibility of materials to microorganisms adhesion was conducted on the basis of results of microbiological tests by ATP (Adenosine 5′-triphosphate) amount measurement (using LuminUltra Photonmaster Luminometer), flow cytometry method (using a Cy Flow Cube 8 cytometer manufactured by Sysmex Partec), and HPC methods with A and R2A agars. Biofilm detachment from the material surface was obtained using a 60-second sonication. Coupons of examined materials were placed in the water supply system for 6 months (December 2017–June 2018).

　　　In order to determine the susceptibility of materials to the formation of biofilms, fractographical parameters were used. The measuring apparatus included Taylor Hobson laser profilometer Talysurf CLI 1000 for fast non-contact measurement of 3D surface topography together with TalyMap and FRAKTAL_Dimmension2D software. The fractal study was conducted on surfaces of selected materials (galvanized steel and PVC) using a box counting method. The method consists of enclosing each section of a profile by a box of width $\varepsilon$ and calculating the area $N(\varepsilon)$ of all of the boxes enclosing the whole profile. This procedure is iterated with boxes of different widths to build a graph; $\ln(N(\varepsilon))/\ln(\varepsilon)$.

　　　In addition, the total height of the roughness profile (Pt) was determined for each profile. The profiled lines with a length of 5 mm separated from the surface of the tested materials were determined with a discretization step of 0.5 μm. The number of profile lines designated on the surface of a given material was 12, which can be considered as adequate [22].

## 3. Results

*3.1. Water Quality Assessment*

　　　Water leaving the experimental system had different parameters in comparison to water supplying it (Table 2). As a result of contact with PVC material, an increase of turbidity and organic compounds content (TOC, oxidizability, UV absorbance) as well as a small increase of nitrites were noted, while ammonium nitrogen content remained unchanged. Simultaneously, content of chlorine (total and free) compounds, dissolved oxygen, and phosphorus compounds decreased. The bacteriological quality of the water also changed; the number of mesophilic and psychrophilic bacteria increased and the presence of *Escherichia coli* (0–48 CFU/100mL) was noted. The presence of bacteria could have been caused by the following: Contamination during assembly, insufficient rinsing, and disinfection of the installation. The high temperature prevailing in the installation (21.53 °C), and the washing out of nutrients from PVC additionally favored the multiplication of microorganisms. Changes to the parameters of water leaving the system pointed to biological processes occurring inside the conduits.

　　　The data presented in Table 3 proves that, in water injected into the experimental system, the admissible content of nutritional substrates N, P, and C was exceeded [23], which could have had a significant influence on the formation of biofilm on the surface of examined materials.

**Table 2.** Inlet and outlet water quality characteristics (N = 26).

| Parameter | Unit | Inlet | | | | Outlet | | | |
|---|---|---|---|---|---|---|---|---|---|
| | | Min | Max | Mean | σ | Min | Max | Mean | σ |
| pH | - | 7.01 | 7.69 | 7.54 | 0.17 | 7.17 | 7.74 | 7.60 | 0.14 |
| Temperature | °C | 14.47 | 20.3 | 17.75 | 1.99 | 10.09 | 23.9 | 21.53 | 3.54 |
| Conductivity | μs/cm | 383 | 506 | 430 | 35.31 | 475 | 662 | 543 | 53.07 |
| Turbidity | NTU | 0.16 | 1.33 | 0.40 | 0.35 | 0.58 | 4.5 | 1.41 | 1.08 |
| Oxidizability | mg $O_2$/L | 0.50 | 2.10 | 1.41 | 0.47 | 0.80 | 2.60 | 1.73 | 0.56 |
| TOC | mg C/L | 0.98 | 2.05 | 1.52 | 0.26 | 1.99 | 5.00 | 2.44 | 0.86 |
| UV absorbance | $UV_{254\,nm}$ | 1.48 | 2.76 | 2.15 | 0.35 | 2.42 | 3.70 | 2.89 | 0.60 |
| Dissolved oxygen | mg $O_2$/L | 12.56 | 16.30 | 14.32 | 1.13 | 5.83 | 10.25 | 9.25 | 1.23 |
| Ammonium nitrogen | mg $N\text{-}NH_4^+$/L | 0.00 | 0.070 | 0.018 | 0.028 | 0.00 | 0.11 | 0.018 | 0.031 |
| Nitrite nitrogen | mg $N\text{-}NO_2^-$/L | 0.00 | 0.037 | 0.003 | 0.010 | 0.001 | 0.037 | 0.0071 | 0.04 |
| Nitrate nitrogen | mg $N\text{-}NO_3^-$/L | 0.09 | 0.90 | 0.49 | 0.29 | 0.20 | 1.50 | 0.52 | 0.376 |
| Phosphates | mg $PO_4^{3-}$/L | 0.02 | 0.19 | 0.053 | 0.047 | 0.00 | 0.15 | 0.038 | 0.037 |
| Total chlorine | mg $Cl_2$/L | 0.01 | 0.21 | 0.102 | 0.07 | 0.01 | 0.07 | 0.027 | 0.017 |
| Free chlorine | mg $Cl_2$/L | 0.01 | 0.08 | 0.033 | 0.02 | 0.00 | 0.04 | 0.012 | 0.011 |
| Mesophilic bacteria (R2A) | CFU/mL | 1 | 100 | 30 | 34 | 300 | 5200 | 2393 | 1561 |
| Psychrophilic bacteria (R2A) | CFU/mL | 5 | 90 | 49 | 32 | 450 | 10600 | 4401 | 3721 |
| *Escherichia coli* | CFU/100mL | 0.00 | 0.00 | 0.00 | 0.00 | 3.00 | 200.00 | 48 | 57.70 |

**Table 3.** Limit values and average concentrations of nutritional substrates in the studied waters (N = 26).

| Stability Criterion | Water Treatment Plant | Inlet (24 h) | Outlet (24 h) |
|---|---|---|---|
| | Mean | | |
| $\Sigma N_{inorg} \leq 0.2$ mg N/L | 0.930 | 0.510 | 0.540 |
| $PO_4^{3-} \leq 0.03$ mg $PO_4^{3-}$/L | 0.027 | 0.053 | 0.038 |
| Dissolved organic carbon (DOC) mg C/L | 2.160 | 1.520 | 2.440 |
| Biodegradable dissolved organic carbon (BDOC) $\leq 0.25$ mg C/L | 0.140 | 0.220 | 0.300 |

### 3.2. Analysis of the Surface of Installation Materials

After six months of contact with tap water, the surface of the examined materials was covered in biofilm. The number of microorganisms populating the material surface varied depending on their type (chemical composition), surface porousness, and method of microbiological estimation (Table 4). Agar R2A stimulated growth of a significantly higher number of microorganisms in comparison to agar A (at incubation times of two and three days), and the obtained results of the two surfaces differed by two orders of magnitude. In the case of agar R2A, both nutrient composition and incubation time (seven days) were different. It should also be noted that the sonication process used during biofilm removal could contribute to damaging part of the microorganisms gathered on the surface of examined materials. A longer incubation period could ensure better conditions for growth of damaged and stressed microorganisms.

Regardless of the method of bacteriological estimation, galvanized steel and PE were the most susceptible to microorganism adhesion materials, followed by chromium-nickel steel and PVC.

The number of mesophilic and psychrophilic microorganisms was comparable for PVC and chromium-nickel steel. In the case of galvanized steel and PE, two times more psychrophilic bacteria than mesophilic bacteria were noted (Table 4). Psychrophilic bacteria are a group of bacteria living and reproducing at low temperatures (0–25 °C) and are mostly Gram-negative G (−). The majority of pathogenic bacteria, as well as ground and water bacteria, belong to mesophilic bacteria [24]. The population size of mesophilic microorganisms provides information on the presence of pathogenic and potentially pathogenic microorganisms, while the population size of psychrophilic bacteria points to organic matter content [25]. Measurements taken via HPC methods, luminometric measurement

of ATP, and flow cytometry cannot be compared to one another with regards to the number of microorganisms due to differences between estimated forms (colonies, ATP, living/dead particles). The obtained data also cannot be compared to values published by other authors, due to differences in the methodology of the conducted experiments (flow and static conditions; different times of biofilm formation, different temperature and medium composition, conditions of removal and estimation of microorganism numbers in the biofilm).

**Table 4.** Total number of microorganisms in the biofilm detachment from the surface of various materials.

| Material | The Number of Microorganisms | | | |
| --- | --- | --- | --- | --- |
| | Agar A (CFU/cm$^2$) | Agar R2A (CFU/cm$^2$) | ATP (RLU/cm$^2$) | Flow Cytometry (Number of Particles/cm$^2$) |
| Galvanised steel | M 35, P 170 | M 9900, P 18,950, | 17,390 | 7,951,795 |
| PE | M 60, P 75 | M 9750, P 18,400 | 8507 | 7,992,750 |
| Chromium-nickel steel | M 53, P 45 | M 5200, P 5800 | 5650 | 7,341,230 |
| PVC | M 80, P 75 | M 3300, P 3200 | 4523 | 7,019,205 |

M—Mesophilic bacteria; P—psychrophilic bacteria; ATP—adenosine 5'-triphosphate; PE—polyethylene; PVC—polyvinyl chloride.

At this stage of our research, we can merely order the materials due to their susceptibility to biofilm formation. Lethola et al. points to copper as the material most resistant to biofilm and states that this material influences biofilm formation solely at the initial stage of contact between water and the material [5]. Copper is often used for in-building systems, the ions of which are toxic to bacteria thus decreasing potential biofilm formation. On the other hand, it should be noted that the material characterizes with high porousness in comparison to plastics, due to which it creates good conditions for colonization of microorganisms [26–28]. In the research [13,18], copper was noted as the least susceptible to biofilm material in comparison to stainless steel and plastics. Our observations suggest that eluting organic matter and biogenic compounds (phosphorus) from the system occurred (Tables 2 and 3) and could stimulate biofilm formation. Yu et al. also points to copper as the most resistant material, followed by PE and galvanized steel [18], which agrees with the order determined by our research: Galvanized steel > PE > chromium-nickel steel > PVC.

The examined chromium-nickel steel characterizes with low surface porousness and high resistance to corrosion. These features could have had a significant influence on the obtained test results. This material characterized with a lower susceptibility to biofilm than steel and PE. However, the main drawback of chromium-nickel is its susceptibility to the presence of chlorine and strong oxidants in water, which translates into formation of pitting corrosion. Protection from this type of undesired situation and the high microbiological quality of water can be achieved through the addition of molybdenum or titan [29]. It is note-worthy that chromium-nickel is a relatively expensive material but, due to its increasing popularity and market competition, its price is systematically decreasing. Surprisingly, PVC turned out to be the most resistant to biofilm material and the cause of this should be examined in further research. Waller et al. conducted research in flow conditions using PVC and cast-iron coupons but the medium circulating in the system was a solution enriched with biogenic substances. Due to the method of biofilm removal from the surface of the examined materials, and more specifically, due to the type of physiological liquid used (phosphate-buffered saline liquid) [19], as well as different media coming into contact with the examined samples, there is no option of direct comparison of obtained test results. Nonetheless, in the case of HPC analysis of agar R2A for the highest number of bacteria, Waller obtained samples from iron (about 8,000,000 (CFU/cm$^2$)) after two days of contact between material and solution, while for PVC, the value of which equaled (80,000 (CFU/cm$^2$)), the samples were obtained after one day.

The results of the fractal analysis of the fractal dimension D values and the total height of the roughness profile together with the error of the standard mean value are summarized (Table 5,

Figures 4–11). The values of both parameters are given as a result of the analysis of the reference material surface and the biofilm material.

**Table 5.** Morphology of the surface of the reference material and material with biofilm.

| Material | Fractographical Parameters | | | |
| | Fractal Dimension D ± Standard Error (-) | | Total Height of the Roughness Profile $P_t$ ± Standard Error (μm) | |
| | For Material | For Material with Biofilm | For Material | For Material with Biofilm |
|---|---|---|---|---|
| Galvanised steel | 1.23 ± 0.016 | 1.18 ± 0.015 | 83.6 ± 6.2 | 393.9 ± 23.4 |
| PE | 1.40 ± 0.015 | 1.41 ± 0.011 | 39.4 ± 2.7 | 93.4 ± 2.9 |
| Chromium-nickel steel | 1.43 ± 0.006 | 1.35 ± 0.006 | 53.6 ± 3.0 | 114.7 ± 4.3 |
| PVC | 1.40 ± 0.029 | 1.39 ± 0.004 | 42.0 ± 1.4 | 30.0 ± 1.3 |

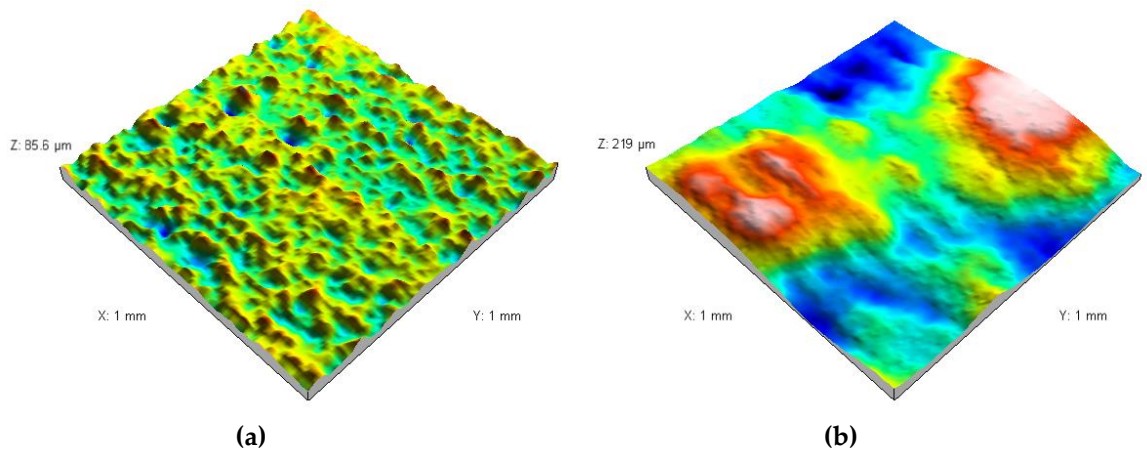

(a)                                                    (b)

**Figure 4.** The surface of the reference material (galvanized steel) (**a**) and material covered with biofilm (**b**).

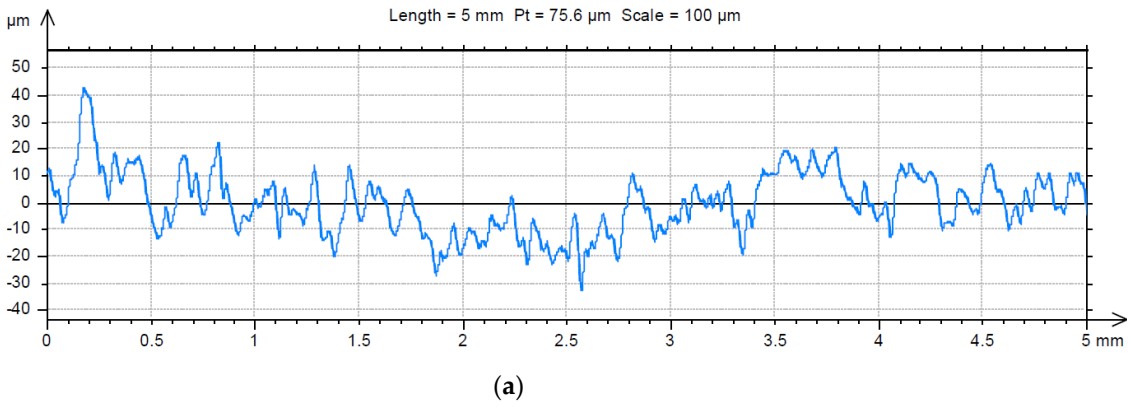

(a)

**Figure 5.** *Cont.*

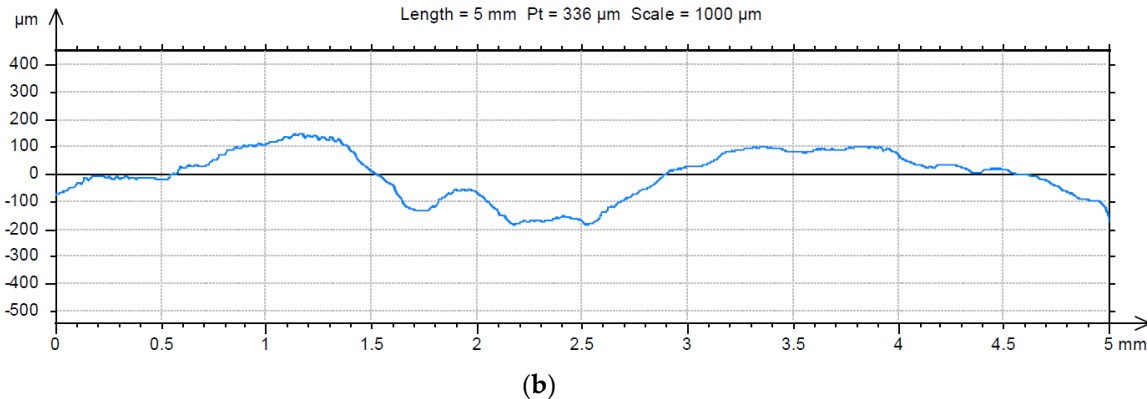

**(b)**

**Figure 5.** Representative profile lines for the reference material (galvanized steel) (**a**) and biofilm coated material (**b**).

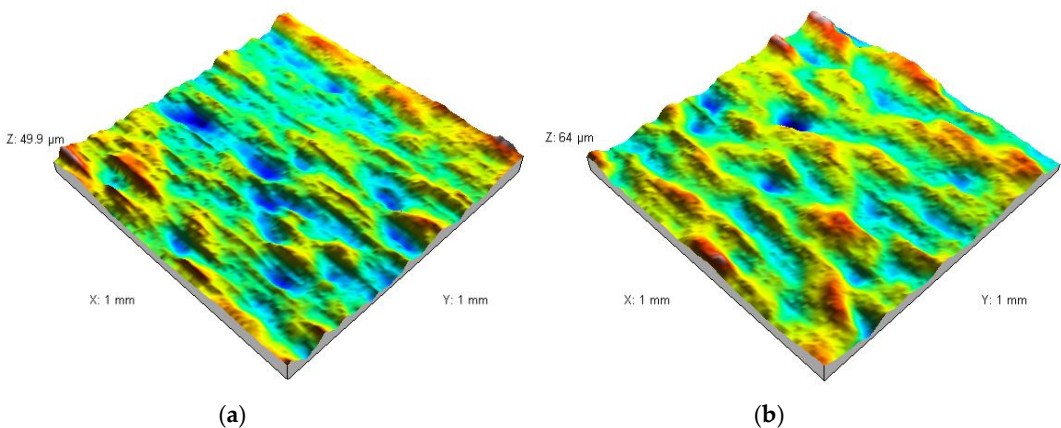

**(a)**　　　　　　　　　　　　　　　　　　　　**(b)**

**Figure 6.** The surface of the reference material (PE) (**a**) and material covered with biofilm (**b**).

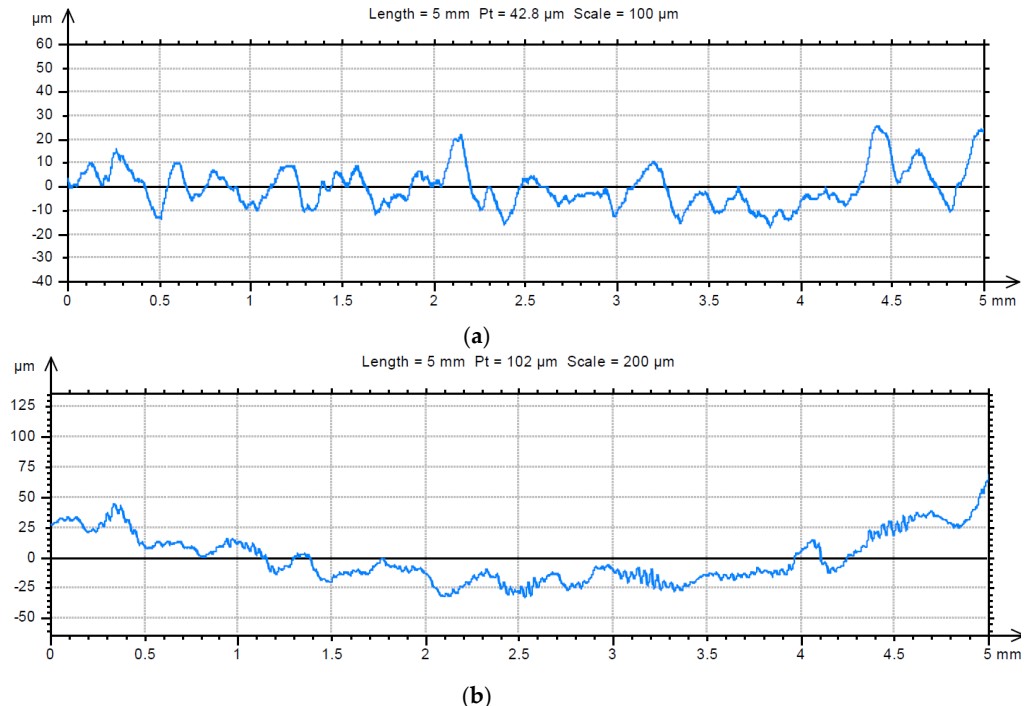

**(a)**

**(b)**

**Figure 7.** Representative profile lines for the reference material (PE) (**a**) and biofilm coated material (**b**).

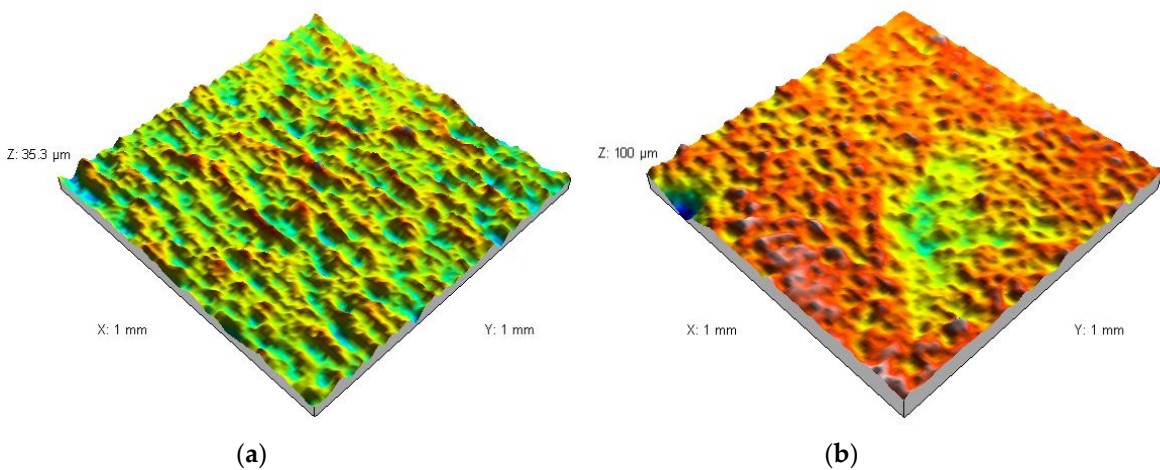

**Figure 8.** The surface of the reference material (chromium-nickel steel) (**a**) and material covered with biofilm (**b**).

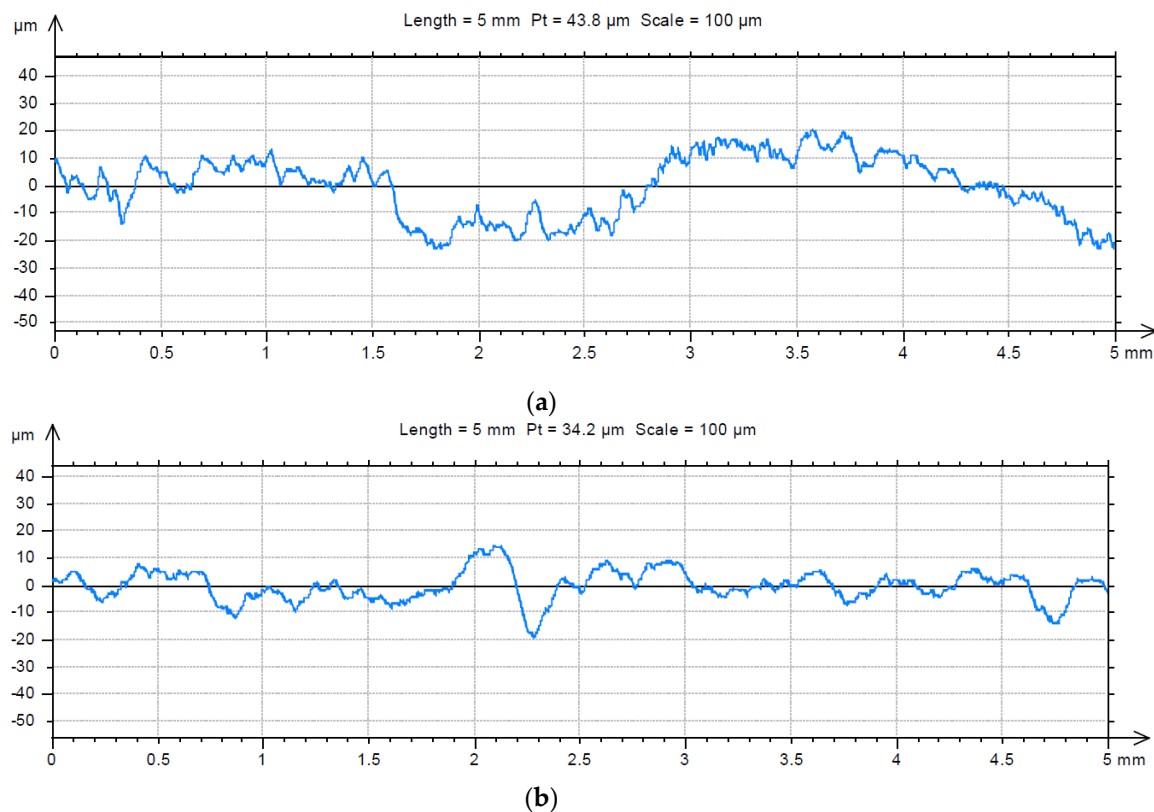

**Figure 9.** Representative profile lines for the reference material (chromium-nickel steel) (**a**) and biofilm coated material (**b**).

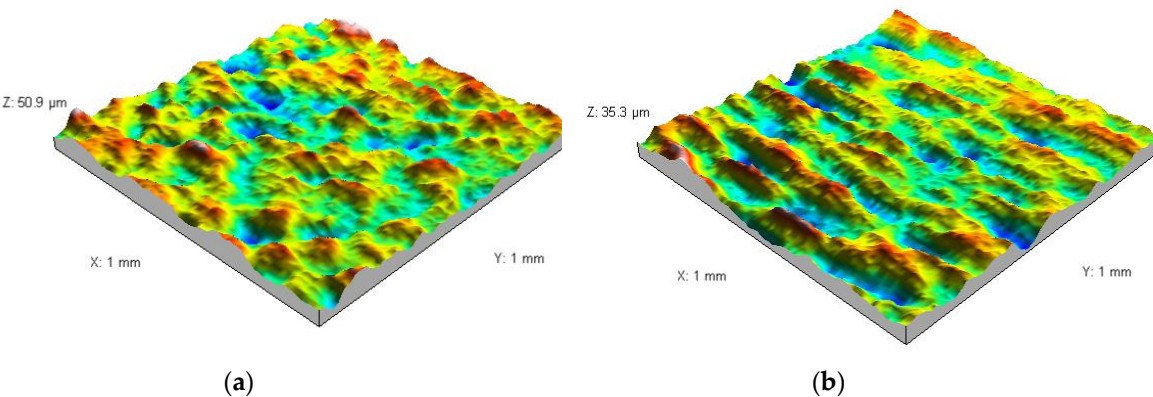

**Figure 10.** The surface of the reference material (PVC) (**a**) and material covered with biofilm (**b**).

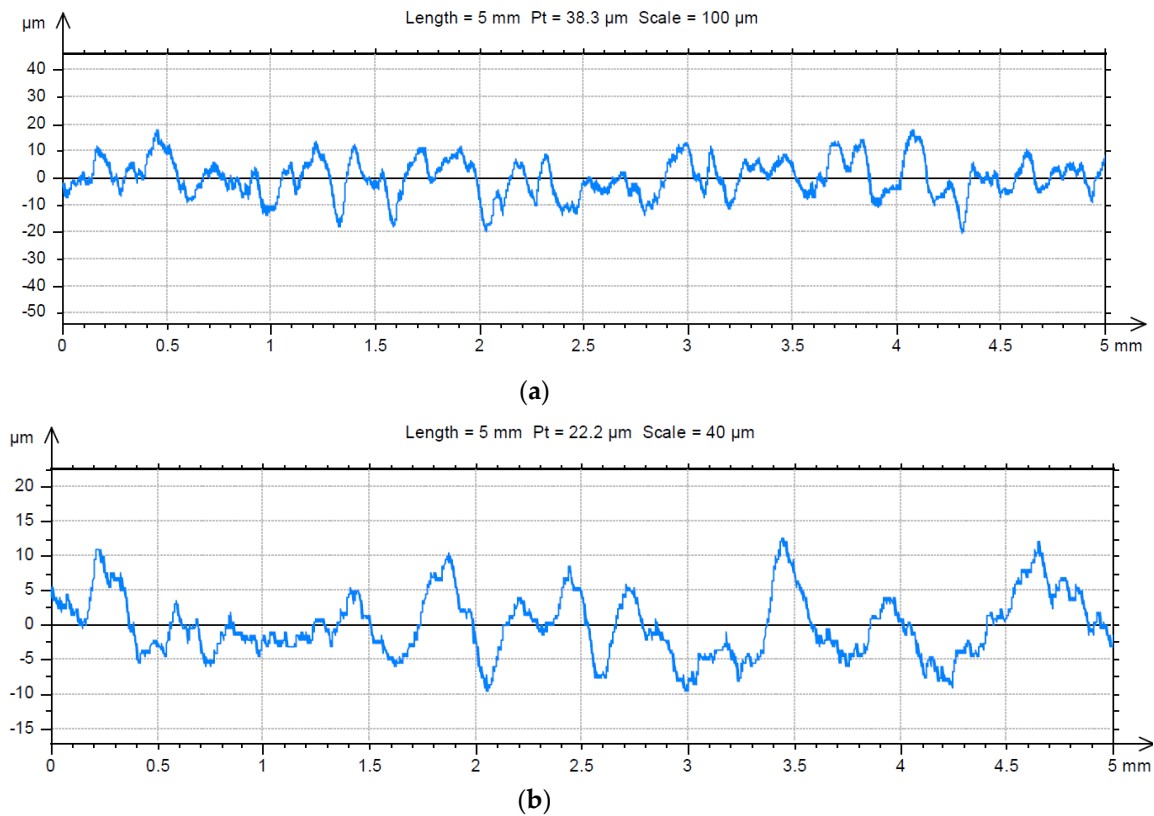

**Figure 11.** Representative profile lines for the reference material (PVC) (**a**) and biofilm coated material (**b**).

## 4. Discussion

In earlier research, Pietrucha et al. confirmed that tap water supplying the experimental system is chemically unstable water with a tendency to solve solids with the possibility of occurrence of slight corrosion [30].

Corrosion encourages biofilm development [31] and presents an additional risk of loss of physical and biological stability of tap water [10]. Striving to achieve biological stability of water directed to distribution networks is connected to the necessity of ensuring extremely low content of nutrients for microorganisms developing on the surfaces of the water-pipe network. This is a very difficult task, especially in the case of water treated in conventional systems (chemical oxidation, coagulation, filtration, disinfection). For the purpose of ensuring biological stability of water, effective elimination of organic substances and biogenic elements, nitrogen and phosphorus, is necessary. In order to maintain

stability and maximally limit the risk of secondary biological water pollution, two out of three biogenes determining microorganism growth should be removed [10]. Zamorska in her research, confirmed the lack of biological stability of water injected into the water-pipe network [32].

In the case of water containing natural organic matter and inorganic nitrogen, phosphorus ions are crucial [2,17]. Too low content hinders microorganism development at a significantly higher degree than in the case of other biogenes [33]. Lehtola et al. suggest that due to the lowest required phosphorus content, it is this element that limits microorganism growth [34]. It should be noted that phosphorus and other nutrients may, in the first days of exploitation of water systems made of plastics, be eluting from these materials, causing quicker development of biofilm, which is what may have occurred in the described case [5,6,18].

Thresholds of parameters limiting redevelopment of microorganisms in distribution networks should be lower than 0.25 mg C/L BDOC, 0.2 mg $N_{norg}$/L and 0.03 mg $PO_4^{3-}$/L [11]. In the first few days after launching the system, concentration of $PO_4^{3-}$ ions increased systematically from 0.05 to 0.15 mg $PO_4^{3-}$/L. No further increase of the analyzed parameter was noted after 26 days (Figure 12). The phosphorus eluted into the water in lower quantities could be used by increased numbers of bacteria developed in the system. Similar research results were noted by [5]. Nonetheless, in conduits made of plastics, small amounts of phosphorus may be present for as long as 200 days [5]. A similar dependency was noted in the case of total organic carbon (TOC), the content of which, after passing through the system, increased from 2.03 mg C/L to 2.93 mg C/L.

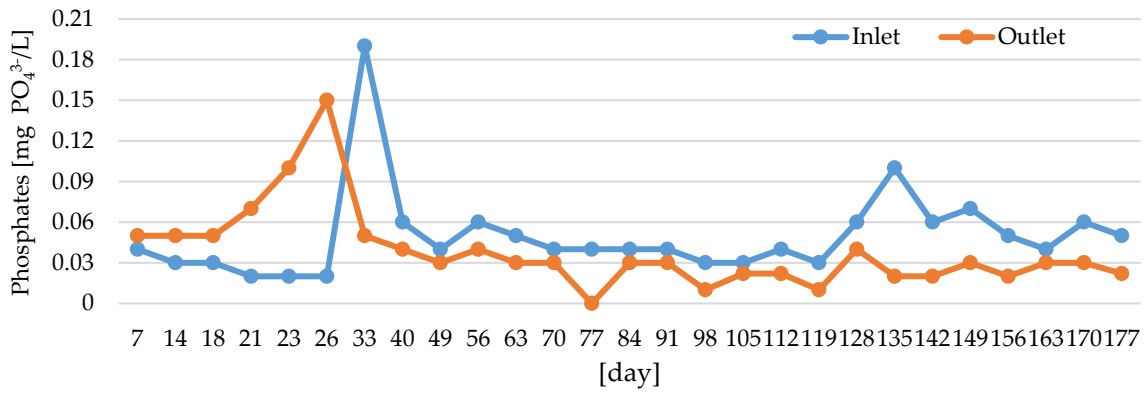

**Figure 12.** Change of phosphate content in the inflow and outflow water from the experimental installation.

Decrease of biofilm growth depends on water temperature, time duration of presence in the system, as well as type and concentration of disinfectant [32,35,36].

Due to the fact that the research was conducted in winter and spring, water temperature had a strong influence on microorganism development. A strong positive correlation between total number of psychrophilic bacteria and temperature was determined [37]. The temperature of water supplying the system changed in the range between 14.6 and 20.3 °C. Additionally, after passing through the system, the water temperature increased maximally up to 24 °C. According to the literary data, microorganism metabolic activity at 7 °C is lower by 50% than at 17 °C [38]. Total and elementary chlorine content in the water leaving the experimental system in comparison to water supplying the system decreased to 0.029 and 0.014 mg $Cl_2$/L, respectively. Chlorine is a disinfectant and a guarantee of the microbiological safety of water, which in this case, was clearly put at risk.

In Poland, the required concentration of elementary chlorine in water injected into the water-pipe network should equal 0.2–0.5 mg $Cl_2$/L, and in water at the ends of the network, it can be no lower than 0.05 g $Cl_2$/L. Gillespie et al. proved that systems distributing water with elementary chlorine concentrations below 0.5 mg $Cl_2$/L were associated with a larger number of bacteria cells in water [39]. Francisque et al. also shows that the number of heterotrophic bacteria was much higher in water samples with chlorine content < 0.3 mg $Cl_2$/L [40].



The proposed new method of assessing the adhesion of microorganisms based on the fractographic analysis of the material surface has proved to be a useful tool in the quantitative description of the biofilm structure.

The obtained values of the fractal dimension D (Table 5) indicate a greater surface roughness of the PVC material than galvanized steel material, which may result in a greater possibility of bacteria deposition in the depressions of this material. As shown by the results of microbiological tests and fractographical analyses, the dominant factor that indicates the possibility of biofilm formation is the type of material, not the roughness of its surface.

The reduction in the fractal dimension of the surface of galvanized steel with biofilm (D = 1.18) compared to the fractal dimension of the surface of this reference material (D = 1.23) indicates the deposition of biological material on this surface. The biofilm stratification is visible on an exemplary scanned fragment of material surface (Figure 4). In addition, the analysis of the shape of the profile lines and the definite change in total height of the roughness profiles indicate a significant occurrence of the biofilm on galvanized steel (Table 5, Figure 5). On the basis of total height of the roughness profile, the thickness of the biofilm layer can be estimated to be around 300 μm (the difference between the total height of the roughness profile of the biofilm material and the reference material).

A similar biofilm formation mechanism was found for chromium-nickel steel. The reduction in the fractal dimension of the surface of chromium-nickel steel with biofilm (D = 1.35) compared to the fractal dimension of the surface of this reference material (D = 1.43) indicates the deposition of biological material on this surface (Table 5, Figure 8). On the basis of total height of the roughness profile, the thickness of the biofilm layer can be estimated to be around 60 μm (Table 5, Figure 9).

In the case of PVC material, the fractal dimension does not show statistically significant (on the significance level of 0.05), changes in the roughness of the profile line. The changes are seen on exemplary fragments of the surface of the material (Figure 10) and in the total height of the roughness profile (Table 5, Figure 11). The reduction of the total height of the roughness profile indicates the formation of a biofilm in the material cavities. As can be seen in Figure 11, the structure is clearly orientated in the direction of the water flow.

No statistically significant difference was also found for PE material. In this case, an increase in total height of the roughness profile was demonstrated (Table 5, Figure 6). The increase in the total height of the roughness profile indicates the deposition of the biofilm not only in the cavities of the material, but also in the creation of new peaks (Figure 7). On the basis of total height of the roughness profile, the thickness of the biofilm layer can be estimated to be around 50 μm (the difference between the total height of the roughness profile of the biofilm material and the reference material).

## 5. Conclusions

Water flowing into the installation did not meet criteria of biological stability. The availability of nutritional substances and an increase in temperature during water residence in the installation resulted in the increase in the number of microorganisms in water and the biofilm formation on internal surfaces of pipes. An increase in turbidity (an average of 1.01 NTU) and decrease in the concentration of chlorine (an average of 74% total chlorine and 63% free chlorine) in water leaving the installation indicated the increase in the risk of the loss of water biological stability and secondary contamination danger.

The development of effective treatment technology, allowing the maintenance of the stability of tap water, and proper selection of installation materials is the key issue in the context of ensuring the health security of consumers. This analysis of obtained results of microbiological tests (ATP, flow cytometry, HTC methods) confirmed that regardless of the material from which the water supply system is made, it is still at risk of formation of biofilm.

Galvanized steel and PE were most susceptible to microorganism adhesion, while PVC and chromium-nickel steel created the least suitable conditions for the development of biofilm (galvanized steel > PE > chromium-nickel steel > PVC).

The most similar amounts of microorganisms occurred on galvanized steel and PE. The biggest differences in the settlement of the surface of materials by microorganisms were found between galvanized steel and PVC. Such a dependence was obtained for all the methods used for the quantitative determination of biofilms. However, due to the high dependence of microbiological determinations on external factors, these conclusions should be confirmed with time-consuming and primary statistical research (ongoing).

**Author Contributions:** Conceptualization, D.P., A.W. and A.P.; methodology, D.P., B.T-C., A.W., J.K, J.Ż and A.D.; validation, D.P., A.D. and J.K.; formal analysis, D.P., A.W. and A.D.; writing—original draft preparation, A.W. and D.P.; writing—review and editing, A.W., D.P., A.D. and J.K.; visualization, A.W., J.Ż. and A.D.; supervision, D.P., B.T-C. and J.K.; project administration, D.P.; funding acquisition, D.P., B.T-C., A and J.K.

**Funding:** This research was funded by subsidies for statutory activity (number: DS.BO.17.001).

**Conflicts of Interest:** The authors declare no conflicts of interest.

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
