# Peer review of "The Impact of the Quality of Tap Water and the Properties of Installation Materials on the Formation of Biofilms"

_water, doi:10.3390/w11091903_

Round 1

Reviewer 1 Report

The aim of the manuscript is to assess changes in the water quality and biological stability depending on the time spent in the distribution system and to determine the susceptibility of materials to adhesion of microorganisms.

I found the study design appropriate, as well as the Methods, which are adequately described. Conversely, the Introduction and the Results sections should be improved by the Authors.

Nevertheless, the manuscript is well organized and interesting, and I think that it would be well received by the readers. For these reasons, I suggest to consider it for the publication after the following major revisions.

1) Abstract section.

The abstract should be better focused on the results obtained. In particular, starting from line 23, the Authors show general results that are difficult to relate to some specific materials used in their experiment. Moreover, I have a major concern with the presence of Escherichia coli in the water, as described in the following "Results section" of this review.

2) Introduction section.

Starting from line 38, it is appreciable that the work takes into consideration the materials, as in the future DIRECTIVE OF THE EUROPEAN PARLIAMENT AND OF THE COUNCIL.

Accordingly, the Authors should improve the Introduction section, making clear reference to the European proposal (especially section 10-bis), both to emphasize the actuality of the article, and to emphasize the importance of the role played by the materials in the quality of the water.

Moreover, it is advisable to also refer to works that describe how the alteration of the water parameters, which can also be attributed to critical issues deriving from anomalies of the distribution networks (ie construction materials), has, for example, led to the request for numerous derogations granted at different European Countries (see for example Azara et al, "Derogation from drinking water quality standards in Italy according to the European Directive 98/83 / EC and the Legislative Decree 31/2001 - a look at a recent past". Ann Ig 2018 ; 30 (6): 517-526 doi: 10.7416 / y.2018.2263). I think that this aspect could be useful for clarification in the choice of investigating the properties of the materials.

3) Results section.

Line 147: The presence of E.coli is extremely critical. The Authors must argue this result (maybe in the Discussion section): where does it come from? Are the supply water controls and the drinking water treatment system really reliable?

Additionally, Table 2 does not show the differences between the materials used: are these differences statistically significant? Are the values compliant with the regulatory limits?

Other minor suggestions are:

line 18: the dot is missing.

line 28: E. coli is an abbreviation. Please, use first the complete name.

line 39: correct the word "biofim"

line 273: are the Authors sure that there is statistical correlation analysis?

Author Response

Thank you for the review, which contributed to improving the substantive level of our article. Thank you for the insight and diligence of the review. Attached, I am sending the corrected article and answers to the review.

Reviewer 2 Report

This study investigated the impact of tap water quality and material type on biofilm formation inside of water conduits. Authors concluded that regardless of material type, biofilm formation is unavoidable thus other ways such as keeping the concentration of key elements low or maintaining a sufficient level of disinfectants in tap water would be more efficient to suppress biofilm formation inside of water conduits.

It is an interesting subject and various analysis methods were used for this study. However, the manuscript needs to be written more concisely and clearly before being considered for publication.

1.       What is the novelty of this study? What new knowledge was found? Authors mentioned previous studies (lines 83-87) testing similar subjects to this study, but it is not clear why another study was needed and what was new or different to the previous studies this time.

2.       Description of experiments (materials and methods) needs more details. For example, how many samples of each kind of material were tested? Did you have an installation (Figure 1) for each test material? This is not written.

3.       (line 124) What are A and R2A agars? What compositions did they have?

4.       (table 2) How many sample were analysed? Values should show errors.

5.       (line 121) section number should be 2.3 not 2.2. (line 142) PCV should be PVC. (line 252) the full name for NOM should be written.

Author Response

Thank you for the review, which contributed to improving and improving the substantive level of our article. Thank you for the insight and diligence of the review. In the attachment I am sending the corrected article and answers to the review.

Reviewer 3 Report

The paper entitled 'The impact of the quality of tap water and the properties of installation materials on the formation of biofilms' aimed to ‘assess changes in water quality and biological stability depending on the time spent in the distribution system and to determine the susceptibility of materials to adhesion of microorganisms'. The researchers suggest that the quality of input water can affect the formation of biofilms and that PVC was the best material to use for preventing biofilm formation.

Introduction – there is a lot of repetition within the introduction and I think it could be made more concise. For example, the following sentences all say that the type of material affects the formation of biofilm/water quality.

Ln37-38: The type of material and the quality of tap water are the most important factors affecting the risk of losing the water safety reaching the consumer.

Ln 42-44: The type of material from which the water conduits are made also has a significant influence on the speed of formation of biofilm, its structure, and biodiversity.

Ln 48-50: Both distribution networks and water supply networks in residential and civic buildings and industrial plants are made of various materials which influence the quality of water reaching the consumer.

Ln 81-82: It is therefore noteworthy that, due to their structure, both plastics and corroding materials (e.g. steel) create different opportunities for formation of biofilm.

In addition, there is reference to several papers stating that the type of material affects either biofilm formation or water quality. More emphasis should be placed on what makes this research unique and different from those studies already mentioned.

Methods – Some information is missing in the methods section. For example,

There are a few abbreviations that need to be written in full (e.g. HTP, HPC, ATP). This also happens in other sections of the article.

Fig 1: At each sample installation point, how were materials assigned; were materials placed in random order? How many replicates of each material were used? Would the water from those nearest the inlet effect the water quality downstream? There are 12 samples installation points prior to disinfection, but only 9 after disinfection. If there were four materials being tested, how were they assigned between these latter 9 sample installation points? Was one material tested at a time in which case would n = 12 and 9 before and after disinfection?

How often was water flushed through the system? Was it every time a sample was taken, leaving the water stagnant between samples, or was it continuous?

The discussion suggests that previous studies have researched the water quality of the inlet water. If so, please include a location of the water treatment works so that this is more obvious.

How many replicate samples were taken and what statistical analysis undertaken?

Table 1: There is no description of incubation temperature and time in Table 1 for the total number of psychrophilic and mesophilic bacteria. Please include this information or a reference. There also needs to be a statement in the table title to say that manufacturer’s instructions were followed unless otherwise stated. Please also include temperature in Table 1.

Was R2A and A agar measuring the same bacteria using two different methods or were psychrophilic and mesophilic bacteria measured using the two different agar. This needs to be expounded on in Table 1.

More information is needed on the ATP measurement; which system and assay was used? Which flow cytometer was used and what method?  Please add more detail or include a reference.

Why was the fractal study used on only two of the materials and not all four?

Results – this article could be significantly improved by providing simple descriptive statistics. For example, n = ?, What method was used to determine confidence intervals? Include confidence intervals in the tables. The results suggest that only mean data is available, but further on in the paper there is reference to time-series data. It would be preferable to display some of the data graphically showing changes over time.

Lines 155-158 should be included in a data analysis sub-section in the Methods section.

Table 3 refers to values that have exceeded limits whilst the limits or where the limits have come from have not been provided.

Where there is more than one image in the figures, they need to be distinguished from each other using lower case letters according to the author guidelines. For examples Figure 4 should read ‘The surface of the reference material (galvanised steel) (a) and material covered with biofilm (b). The same goes for Figures 5, 6 and 7.

Discussion and Conclusions – Conclusions are based on data that has not been analysed statistically. For example, in Table 4 the number of bacteria measured using R2A agar looks very similar between galvanised steel and PE. A simple T-test (or the non-parametric equivalent depending on how the data is distributed) would give us a greater understanding of how different these values actually are. Another example is in Ln2 64-265: ‘Phosphorus content in the water leaving the PE pipes was significantly higher in the first few days of the experiment (5-7μg/l) in comparison to water supplying the system (2 μg/l), stabilising after about 20 days.’ What statistics were used to base this conclusion on and can the P value be included in the text. Until statistics have been completed on the data, it is not possible to make conclusions about the data in the rest of the article.

The discussion also includes comments about data that is not shown in the article. For example, ‘In the first few days after launching the system, concentration of PO43- ions increased systematically from 0.05 to 0.15 mg PO43-/L.’ As has been mentioned previously, there is no time series data recorded in the results section.

It is possible that with some tweaking of the manuscript, an emphasis on how this research is unique and with the inclusion of statistics that this will show some interesting data and the authors should be encouraged to work on this further.

Author Response

Thank you for the review, which contributed to improving and improving the substantive level of our article. Thank you for the insight and diligence of the review. In the attachment I am sending the corrected article and answers to the review.

After the review, the manuscript was thoroughly checked and corrected, and all missing elements were completed in accordance with the recommendations of the reviewers.

Round 2

Reviewer 1 Report

I believe the Authors' answers are comprehensive and have addressed all the raised issues.

For these reasons, the manuscript is now publishable on Water.

Author Response

Once again, thank you very much for preparing the review of our article.

Reviewer 2 Report

Questions were answered sufficiently, the revised manuscript has improved. 

Author Response

(The authors gave the same response as above.)

Reviewer 3 Report

Whilst I agree that this article is of interest to the wider scientific community, I feel that the authors should have waited until they have carried out "time-consuming" statistical analysis on their data before submitting. There is data missing from the manuscript including phosphorous time-series data, and fractal data was only shown for two of the four materials tested. When challenged the authors argue that "the graphical representation of changes in phosphorus content was abandoned because then it would also be necessary to present changes to other biogenic substances (nitrogen and carbon). This will be discussed in another scientific article". and that "Due to the large volume of the article, the two most representative samples were selected. The remaining results will be published in the next scientific article." As well as contravening the author guidelines for the Water journal "Water has no restrictions on the length of manuscripts, provided that the text is concise and comprehensive." and "Water requires that authors publish all experimental controls and make full datasets available where possible" it also suggests that the authors are splitting the data either in order to get more publications or because they haven't yet completed the statistical analysis. I would advise the authors to do the stats and provide the full datasets, using the Supplementary Material if necessary and resubmit a much more thorough evaluation of the research that they have carried out. It is impossible to make assumptions about the data without first having verified it through statistical analysis. I would look forward to reviewing this manuscript when submitted in full.

Author Response

(The authors gave the same response as above.)
